# Physical and mental health well-being of COVID-19 recovered patients: A phenomenological study

Sawsan Abuhammad[1]*, Karem H. Alzoubi[2], Omar F. Khabour[3], Shaher Hamaideh[4], Basheer Khasawneh[5], Maryam El-zubi[6]

**1** Department of Maternal and Child Health, Faculty of Nursing, Jordan University of Science and Technology, Irbid, Jordan, **2** Department of Clinical Pharmacy, Jordan University of Science and Technology, Irbid, Jordan, **3** Department. of Medical Laboratory Sciences, Jordan University of Science and Technology, Irbid, Jordan, **4** Department of Community and Mental Health Nursing, Faculty of Nursing, The Hashemite University, Zarqa, Jordan, **5** Department of Internal Medicine, Faculty of Medicine, Jordan University of Science and Technology, Irbid, Jordan, **6** Department of Pharmacy Practice, College of Pharmacy, Gulf Medical University, Ajman, United Arab Emirates

* Shabuhammad@just.edu.jo

## Abstract

### Aim

This study aims to describe the experience of COVID-19 recovered patients' physical and mental health well-being.

### Method

A qualitative research approach was employed utilizing an **u**nstructured interview protocol **t**o explore the experiences of individuals recovering from COVID-19. Data were collected from 30 participants who had recovered from a moderate to severe form of the disease, all of whom required hospitalization with oxygen support during their illness. To gain an in-depth understanding of their post-recovery experiences, grand tour questions were used to facilitate open-ended discussions, allowing participants to fully articulate their perspectives on the impact of COVID-19 on their health and well-being.

### Results

Thematic analysis of the interviews revealed four primary themes related to the post-recovery experiences of COVID-19 patients. Physical health concerns were widely reported, including persistent respiratory difficulties, joint and muscle pain, changes in activity levels, and the worsening of pre-existing health conditions. In terms of cognitive health, participants described experiencing memory loss, difficulty concentrating, and other cognitive impairments that affected their daily functioning. Psychological health challenges were also prominent, with many participants

**Data availability statement:** All relevant quantitative and coded qualitative data used to support the findings of this study are included within the paper.

**Funding:** This study was supported by a grant (MPH/01/14/2021) from the Scientific Research Funds, Ministry of Higher Education and Scientific Research, Amman, Jordan.

**Competing interests:** The authors have declared that no competing interests exist.

expressing feelings of anxiety, nervousness, loneliness, and sadness, reflecting the emotional toll of their illness and recovery. Additionally, sleep disturbances emerged as a significant issue, with individuals reporting difficulty falling asleep, fragmented sleep patterns, and persistent fatigue. These findings indicate that COVID-19 recovery extends beyond physical healing, affecting multiple aspects of an individual's overall well-being.

## Conclusion

This study highlights the extensive and multidimensional impact of COVID-19 recovery, affecting physical, cognitive, and psychological health, as well as sleep patterns. The persistence of symptoms such as respiratory issues, cognitive impairments, emotional distress, and sleep disturbances underscores the need for long-term medical and psychological support for recovered patients. These findings emphasize the importance of comprehensive post-recovery **care**, including rehabilitation programs, cognitive interventions, and mental health services to support individuals in regaining their overall well-being. Future research should focus on developing targeted interventions to address these long-term effects and improve the quality of life for COVID-19 survivors.

---

### Introduction

The emergence of severe COVID-19 infection has underscored the profound and enduring impact of the virus on individuals who undergo critical illness and recovery [1–3]. As the global community addresses the repercussions of the pandemic, it is imperative to examine the long-term physical and psychological health trajectories of individuals who have experienced severe cases of COVID-19 [4–6]. Investigating the health outcomes following severe infection is not merely an academic pursuit but a crucial initiative aimed at understanding, addressing, and mitigating the complex and persistent challenges faced by survivors [2,7].

Beyond the immediate effects of acute illness, survivors of severe COVID-19 often experience a broad spectrum of physical and psychological health consequences that warrant significant attention [8,9]. These effects extend beyond hospitalization, encompassing rehabilitation, mental health support, and long-term care [6,10]. This study aims to provide a comprehensive analysis of post-severe infection health, with the objective of enhancing the understanding of the recovery process while informing the development of targeted interventions, specialized healthcare strategies, and an integrated framework for evaluating factors that influence the long-term well-being of survivors [6,8].

The scope of this investigation extends beyond the assessment of physical symptoms and psychological challenges; it seeks to cultivate a comprehensive understanding that enables healthcare providers, policymakers, and communities to implement tailored and effective support mechanisms [11]. By examining the physical

and mental health of individuals who have endured severe COVID-19, this study endeavors to improve survivors' quality of life, reduce long-term health risks, and contribute to a more informed and compassionate response to the post-infection consequences of severe COVID-19. A substantial proportion of COVID-19 survivors experience persistent physical symptoms, commonly classified as post-COVID-19 syndrome [7,12]. Investigating these long-term health implications is essential for identifying specific medical concerns that may require targeted interventions and sustained clinical management [13].

The psychological consequences of the pandemic are significant, with many survivors experiencing mental health challenges [14,15]. Examining these aspects is essential for implementing appropriate psychological interventions and mental health support services that address the emotional well-being of individuals who have endured the physical and psychological strain of COVID-19 [16]. Furthermore, understanding the long-term health consequences of COVID-19 is essential for optimizing healthcare resource allocation [17]. This knowledge enables healthcare systems to anticipate and address the evolving needs of survivors, ensuring that medical services are appropriately designed to support various aspects of recovery [18,19]. Public awareness of post-COVID-19 health outcomes plays a pivotal role in shaping public health policies, vaccination strategies, and preventive measures that account for the potential long-term effects of the virus on individuals and communities [20].This study aims to explore and describe the experiences of individuals recovering from COVID-19, focusing on their physical and mental health well-being.

## Method

In this study, a qualitative approach was adopted using the unstructured interview protocol. The unstructured interviews were conducted with COVID-19 recovered patients to explore their lived experiences regarding physical and mental health well-being. Participants shared personal narratives detailing their journey from illness to recovery, shedding light on lingering physical symptoms, emotional distress, and the broader social and lifestyle impacts of post-COVID health challenges. Through open-ended discussions, valuable insights were gathered on the complexities of long-term recovery, including the need for targeted medical support, mental health care, and social understanding. These interviews provided a rich, in-depth understanding of survivors' struggles and resilience, offering a foundation for further analysis and recommendations for healthcare interventions. Data were collected from 30 COVID-19 recovered patients who experienced moderate to severe form of the disease (hospitalized with oxygen support). Regarding sample size, the authors determined that 30 participants would be sufficient based on the concept of data saturation, which is widely accepted in qualitative research as the point at which no new themes or insights emerge from the data. Grand tour questions were used during the interviews to fully describe this phenomenon among COVID-19 recovered patients. Unstructured interviews opened the door widely for experience suffering from COVID-19 virus perspectives and allow the researcher to probe to get a complete picture of the issue. Furthermore, qualitative interviews may give some extended statements in which the researcher can make some comparative issues between these statements and the literature review.

### Setting

The study was conducted among COVID-19 recovered patients at their homes or at an agreed meeting place based on their preferences.

### Data collections

The data collection was as the following. 1. Official approval from the hospital directorate. 2. Requesting the list of COVID-19 recovered patients from the hospitals. The list was used to select individuals who satisfied the selection procedure. These criteria include experiencing a moderate to severe case of COVID-19 that required hospitalization, being over 18 years of age, and possessing the ability to speak and read the Arabic language. 3. Conducting one-on-one interviews with prospective respondents to introduce the investigator, explain the study's objective, and study procedure. There was a

discussion of the respondent's right to confidentiality and their withdrawal at their convenience. 4. All required information was explained to the participants. 5. If the individual agrees to participate; they signed the consent form. Arrangement for time and location for interview was made. (The COVID-19 recovered patients at was interviewed according to the preference. 6. Each interview lasted for around 45 minutes to one-hour length 7. Tape-recording and verbatim transcription of the interview was done. Then, translating meaningful quotes into English followed. Back translation and transcribed text comparison were used to achieve translation verification. 8. A follow-up phone call was utilized in exploring new arising issues. The data collection was between 1st of August 2023 and 30th of November 2023

## Interview procedure

This protocol was designed to facilitate an open and flexible dialogue, allowing participants to share their experiences in their own words without being confined to predetermined questions.

The unstructured interview process focused on creating a conversational environment where participants could freely discuss their physical and mental health challenges post-recovery. The interviews began with a broad, open-ended prompt, such as:

• **"Can you describe your experience recovering from COVID-19?"**

From this initial prompt, the interview progressed organically, with follow-up questions tailored to each participants' responses. The interviewer actively listened and probed deeper based on emerging themes, ensuring that both physical and mental health aspects were explored. Common areas of discussion included:

1. **Physical Health Experiences**

   ◦ Persistent symptoms (e.g., fatigue, breathing difficulties, joint pain)

   ◦ Changes in physical abilities or endurance

   ◦ Need for ongoing medical care or rehabilitation

2. **Mental and Emotional Well-being**

   ◦ Psychological impact of COVID-19 recovery

   ◦ Anxiety, depression, or stress related to post-recovery challenges

   ◦ Social and emotional support systems

3. **Healthcare and Support Systems**

   ◦ Access to medical care and follow-up treatment

   ◦ Perceptions of healthcare support during and after recovery

   ◦ Recommendations for improving post-COVID care

The unstructured nature of the interviews allowed participants to express their experiences holistically, uncovering rich, in-depth insights that align with the phenomenological approach. The collected narratives provided a comprehensive understanding of the complex and individualized nature of post-COVID recovery.

## Ethical considerations

This study approved from Jordan University of Science and Technology IRB (# 70/2021).It was according to declaration of Helsinki. Written informed consent was signed from each participant in the study. In our study, participants were not

compensated for their participation. This decision was based on the university IRB. All participants were fully informed about the voluntary nature of their involvement and provided written informed consent prior to data collection.

## Data analysis

The collected data were analyzed using content analysis, with a systematic approach to organizing and coding the data to identify recurring patterns. Themes were identified based on similarities, and categories were developed accordingly. Each transcript from COVID-19 recovered patients was thoroughly reviewed and analyzed following Berland's framework (53). This framework consists of seven steps for data analysis: (1) transcribing the interviews, (2) comprehending the content, (3) coding the data, (4) developing an analytical framework, (5) applying the framework to the data, (6) integrating data into the matrix framework, and (7) interpreting the findings. The authors used NVivo 12 primarily for coding and data organization. The software facilitated the systematic categorization of interview transcripts, enabling the identification of themes and subthemes through both inductive and deductive approaches. It also supported data management by allowing the authors to easily retrieve coded segments, generate coding matrices, and visualize theme interrelations.

The trustworthiness of qualitative research is established through credibility, dependability, confirmability, and transferability. In this study, the credibility (validity) of the findings was ensured through two approaches. First, the researcher randomly selected five participants and provided them with a summary of their shared experiences. Participants were asked to verify whether the findings accurately reflected their experiences as discussed during the interview. Agreement among participants confirmed the accuracy of the data interpretations. Second, the findings were validated through cross-checking with two independent expert researchers with doctoral degrees. These independent analysts conducted their own analysis of all interviews, and their identified themes were compared with those derived by the original researcher. Consensus among all analysts confirmed the reliability and validity of the study's findings.

## Results

### Demographic variables

The following table shows the characteristics of the participants. The participants were consisted of 10 male (33%) and 20 females (66%). See Table 1

### Themes

There are four themes that emerged from the interviews. These themes are physical health, cognitive health, psychological health, and sleeping problems.

### Respiratory problems

A majority of the participants (22 out of 30; **73.3%**) reported experiencing respiratory issues after recovering from COVID-19. Examples include:

• **F (75):** "I feel suffocation, shortness of breath, and difficulty sleeping all night because of my breathing."

• **F (48):** "I feel short of breath, but most of all when I sleep."

• **M (52):** "My condition is worse than those who smoke. My chest is ruined by the coronavirus; I cannot take a breath."

### Joint and muscle pain

A significant portion of participants (18 out of 30; **60%**) reported persistent joint and muscle pain post-recovery. Some participants stated:

**Table 1. Characteristics of the participants (N = 30).**

| Item | | Frequency | Percent |
|---|---|---|---|
| What is your gender? | Female | 20 | 66.7 |
| | Male | 10 | 33.3 |
| What is your current employment status? | Student | 1 | 3.3 |
| | Freelance | 1 | 3.3 |
| | Not work | 14 | 46.7 |
| | Retired | 8 | 26.7 |
| | Employee | 6 | 20.0 |
| What is your monthly income in dinars? | 401 to 800 JOD | 13 | 43.3 |
| | Less than 400 JOD | 15 | 50.0 |
| | More than 1500 | 2 | 6.7 |
| What is your level of education? | Primary and Secondary | 16 | 53.3 |
| | Bachelor | 5 | 16.7 |
| | Diploma | 5 | 16.7 |
| | Graduate | 3 | 10.0 |
| | University student | 1 | 3.3 |
| What is your marital status? | Single | 1 | 3.3 |
| | Married | 29 | 96.7 |
| Where do you live? | Village | 11 | 36.7 |
| | City | 19 | 63.3 |
| Do you smoke cigarettes or hookah? | No | 28 | 93.3 |
| | Yes | 2 | 6.7 |
| Have you been infected with Corona more than once? | No | 21 | 70.0 |
| | Yes 3 times | 5 | 16.7 |
| | Yes, two times | 4 | 13.3 |
| What kind of vaccine did you receive? | AstraZeneca | 2 | 6.7 |
| | Chinese | 12 | 40.0 |
| | Pfizer | 13 | 43.3 |
| | I have not received any vaccine | 3 | 10.0 |
| Since when you first got infected with the Corona virus? | 18-24months | 11 | 36.7 |
| | more than24months | 19 | 63.3 |
| Since when you got infected with the Corona virus the second time (if you were infected more than once)? | | 17 | 56.7 |
| | 12-18months | 1 | 3.3 |
| | 18-24months | 2 | 6.7 |
| | more than 24months | 9 | 30.0 |
| | Mild | 1 | 3.3 |
| How would you describe your infection with Corona? | Mild at first time | 3 | 10.0 |
| | Mild at second time | 6 | 20.0 |
| | I just affected one time and it was mild | 1 | 3.3 |
| | I just affected one time and it was severed | 19 | 63.3 |
| | Equal both times | 1 | 3.3 |
| Which of the following symptoms appeared on you during your infection with Corona [high temperature]] | No | 1 | 3.3 |
| | Yes | 29 | 96.7 |
| Which of the following symptoms appeared during your infection with Corona [pain in the throat] | No | 2 | 6.7 |
| | Yes | 28 | 93.3 |

*(Continued)*

| Item | | Frequency | Percent |
|---|---|---|---|
| Which of the following symptoms did you have during your infection with Corona [cough] | No | 6 | 20.0 |
| | Yes | 24 | 80.0 |
| Which of the following symptoms did you have during your infection with Corona [difficulty breathing]] | No | 1 | 3.3 |
| | Yes | 29 | 96.7 |
| Which of the following symptoms appeared during your infection with Corona [lack of oxygen]] | No | 5 | 16.7 |
| | Yes | 25 | 83.3 |
| Have you been admitted to the hospital? | No | 10 | 33.3 |
| | Yes | 20 | 66.7 |
| Did you need oxygen? | No | 6 | 20.0 |
| | Yes | 24 | 80.0 |
| Have you been admitted to intensive care? | No | 19 | 63.3 |
| | Yes | 11 | 36.7 |
| Which of the following health problems persisted for a period of time after you recovered from Corona [feeling short of breath] | No | 5 | 16.7 |
| | Yes | 25 | 83.3 |
| Which of the following health problems persisted for a period of time after you recovered from Corona [chest pain] | No | 15 | 50.0 |
| | Yes | 15 | 50.0 |
| Which of the following health problems persisted for a period of time after you recovered from Corona [feeling tired] | No | 2 | 6.7 |
| | Yes | 28 | 93.3 |
| Which of the following health problems persisted for a period of time after you recovered from Corona [cough] | No | 18 | 60.0 |
| | Yes | 12 | 40.0 |
| Which of the following health problems persisted for a period of time after you recovered from corona [memory loss]] | No | 7 | 23.3 |
| | Yes | 23 | 76.7 |
| Which of the following health problems persisted for a period of time after you recovered from corona [delirium]] | No | 26 | 86.7 |
| | Yes | 4 | 13.3 |
| Which of the following health problems persisted for a period of time after you recovered from Corona [feeling depressed] | No | 8 | 26.7 |
| | Yes | 22 | 73.3 |
| Which of the following health problems persisted for a period of time after you recovered from Corona [feeling anxious] | No | 3 | 10.0 |
| | Yes | 27 | 90.0 |
| Which of the following health problems persisted for a period of time after you recovered from Corona [repeated diarrhea] | No | 19 | 63.3 |
| | Yes | 11 | 36.7 |
| Which of the following health problems persisted for a period of time after you recovered from Corona [feeling tired]] | No | 3 | 10.0 |
| | Yes | 27 | 90.0 |
| Which of the following health problems persisted for a period of time after you recovered from corona [sleep disorders]] | No | 1 | 3.3 |
| | Yes | 29 | 96.7 |
| Which of the following health problems persisted for a period of time after you recovered from corona [joint pain]] | No | 3 | 10.0 |
| | Yes | 27 | 90.0 |
| Which of the following health problems persisted for a period of time after you recovered from corona [muscle pain] | No | 2 | 6.7 |
| | Yes | 28 | 93.3 |

- **M (63):** "I always complain of muscle and joint pain."

- **M (52):** "I always feel tired in the muscles and joints and pain in the whole body. When you put your hands on the meat, it becomes painful in its place. The whole body feels as if it is weak. My hands and running all hurt. I go and take injections until the pain subsides."

- **F (53):** "I have pain in my joints and muscles. I don't know if it is from the joints or from the coronavirus. My problem is worsening, no medication could help me decrease my pain."

### Change in activity level

A considerable number of participants (17 out of 30; **56.7%**) reported changes in their activity level, stating that they became fatigued even with minor physical exertion. Some accounts include:

- **F (49):** "When I walk, go up a flight of stairs, or even sometimes when I talk or do any simple effort, I feel tired. I have felt this way for almost a whole year, and I keep returning to the hospital because of it."

- **F (48):** "The feeling of fatigue is still present. Before COVID-19, I was very active, but now, if I do anything quickly, I get tired and have to rest."

- **M (65):** "If I want to go up the stairs, I take one step, stop, relax, and then continue. I cannot keep going at a normal pace."

### Worsening of physical problems

Half of the participants (15 out of 30; **50%**) indicated that their pre-existing physical issues worsened post-COVID-19. For example:

- **F (75):** "My ears became blocked, and I stopped hearing well. The doctor told me that my ear nerve had become weak. The problem existed before COVID-19 but has worsened significantly."

- **F (41):** "My body can no longer tolerate any illness. Even a simple flu makes me feel extremely weak."

- **F (75):** "I had joint pain before, but after COVID-19, the pain increased so much that I can no longer walk without assistance."

- **F (53):** "I had a disc problem before, but post-COVID, the pain has intensified in my joints, fingers, and lower back."

### Cognitive health

**Loss of memory and diminished ability to focus.** A total of 12 participants (12 out of 30; **40%**) reported issues with memory loss and concentration difficulties. Some stated:

- **F (58):** "I have lost some memory, and my memorization is slower. I used to memorize the Quran with ease, but now it is more difficult."

- **F (58):** "I sometimes go to get something and forget what I was looking for. I even search for my phone while I am holding it."

- **M (57):** "I forget a lot. When someone passes in front of me and speaks, I forget their appearance even if I remember their name. My vitamin B12 and D levels are normal, but my memory has deteriorated."

**Change in cognitive abilities.** About one-third of the participants (10 out of 30; **33.3%**) reported changes in cognitive function. Some examples include:

- **M (55):** "I used to have a very strong memory, but after COVID-19, I feel like I have forgotten so much, and my memory has declined significantly."

- **M (58):** "I sometimes experienced delirium. I would talk about things that never happened, but thankfully, I have improved over time."

## Psychological health

**Anxiety and nervousness.** More than half of the participants (16 out of 30; **53.3%**) reported increased anxiety and nervousness post-COVID-19. Some examples include:

• **F (58):** "I quickly became nervous. Even my children notice and ask, 'What's wrong with you, Mama?'"

• **F (69):** "I constantly feel anxious, especially at night."

• **F (68):** "The impact of COVID-19 on my body is huge. It has also significantly affected my sleep."

• **F (61):** "I have remained nervous. Life and its struggles weigh on me. I even started wishing that I had died like those who passed away from COVID-19."

**Feeling of loneliness and sadness.** A significant portion of the participants (14 out of 30; **46.7%**) reported feeling lonely and sad. Some accounts include:

• **M (73):** "I always feel sad and depressed because I feel like a stranger to my own sons and daughters."

• **M (52):** "I prefer to be alone and isolated. Sometimes, I feel deeply sad and depressed."

## Sleeping problems

Almost all participants (28 out of 30; **93.3%**) reported experiencing sleeping disturbances after COVID-19. Some reported:

• **F (69):** "I do not sleep well. I keep thinking about things that are not there. My sleep is intermittent and never deep."

• **M (65):** "After COVID-19, my sleep is interrupted. I sleep for just an hour or two and struggle to fall back asleep. Sometimes, I stay awake for hours before I can sleep again."

• **F (68):** "I have trouble sleeping. If I wake up at night, I cannot get back to sleep."

• **F (45):** "I do not sleep at all. Even when I do, it is light and not deep sleep. I feel very anxious. Sometimes I consider visiting a doctor to prescribe sleep aids. I stay on my phone to make myself drowsy."

## Discussion

This qualitative study delved into the recovery journey of individuals diagnosed with COVID-19 and required admission, aiming to enhance our quantitative comprehension of post-discharge results. Our results affirm these individuals enduring physical and psychological symptoms. Through this study, the authors revealed that these individuals in our study encountered a shift in their perceptions of their bodies.

The first theme that emerged from our study is physical health that include respiratory problems, joint and muscle pain, changes in activity level, and worsening of previous health problems. Similarly, deterioration in physical well-being was primarily observed in aspects such as physical role functioning, mobility, and regular activities, which are linked to engagement in work and everyday tasks [21,22]. Furthermore, a decrease in physical capacity was evident, as indicated by post-COVID- 19 results being lower than the typical average of >650 meters in healthy adults [23,24]. In a study, muscle strength was assessed three months after recovering from COVID-19, revealing that 18% of individuals who were in the ICU developed weakness [25]. Survivors of COVID-19 have also been observed to experience muscle weakness and acute sarcopenia [26]. Paneroni et al. discovered quadriceps weakness in 86% and biceps brachialis weakness in 73% of individuals infected with the coronavirus upon hospital discharge [27]. This underscores the significance of assessing muscle strength during follow-up [28,29].

Many participants in our study mentioned decreased of physical functions. In agreement to our study, studies showed that both patients who were hospitalized (in ward and ICU settings) and those who were not hospitalized exhibited a decline in function and decreased physical capacity during the 1- and 3-month follow-up periods after recovering from COVID-19 [30,31]. Consequently, the reduced physical capacity and function observed in individuals recovering from COVID-19 may also result from physical inactivity resulting from quarantine, social distancing, and isolation measures.

In our study, many participants reported experiencing mental health challenges, including symptoms commonly associated with anxiety and depression. However, it is important to note that these symptoms were self-reported and not formally diagnosed or measured using validated assessment tools. Instead, participants described feelings of persistent worry, nervousness, sadness, and emotional distress, which align with anxiety-depressive symptoms. These findings highlight the need for further clinical evaluation and support for individuals recovering from COVID-19. Similarly, many studies reviewed that reported anxiety symptoms were identified as the predominant mental issue commonly observed in post-COVID-19, with prevalence rates ranging from 7% to 64% [4,32–34]. Reported depression symptoms emerged as the second most prevalent psychological issue, with rates varying between 4% and 31% for periods exceeding one-month post-recovery. Upon hospital discharge, both depression and anxiety symptoms were reported by 32% of patients in Italy and 41% in Iran. Another notable post-COVID-19 mental health issue was symptoms similar to PTSD, with rates ranging from 13% to 47%. Many individuals described experiencing intrusive thoughts, flashbacks of their hospitalization, and heightened anxiety related to their illness and recovery. Some reported persistent fear of reinfection, distressing nightmares, and and an exaggerated startle response. These self-reported PTSD-related symptoms suggest a profound psychological impact of COVID-19, emphasizing the need for targeted mental health interventions and long-term psychological support for recovered individuals. Sleep problems were also prevalent, with rates between 18% and 31%, and cognitive-functional issues were reported in 17% to 4% of post-COVID-19 individuals, particularly among ICU survivors [11,35].

In our study, many participants mentioned the impact of disease on their memory. Accordingly in previous studies, memory were moderately decreased in 58% and severely decreased in 19% of COVID-19 survivors [36]. Many participants suffered from mental health issues after recovering from the infection [37]. Sleep difficulties, especially insomnia, were identified as as a significant mental health challenge following the COVID-19 pandemic, manifesting in both short-term and long-term phases [38]. The physiological and psychological stress during the acute phase of the infection were proposed to activate mechanisms leading to the release of pro-inflammatory cytokines, disrupting metabolic and cardiovascular functions associated with sleep [39]. Researchers suggested that sleep issues may negatively impact emotional and attentional regulation, resulting in irritability and difficulties concentrating. Additionally, other studies highlighted sleep problems as a primary complication reported by individuals recovering from COVID-19 [40,41].

## Conclusion

This study highlights the extensive and multidimensional impact of COVID-19 recovery, affecting physical, cognitive, and psychological health, as well as sleep patterns. The persistence of symptoms such as respiratory issues, cognitive impairments, emotional distress, and sleep disturbances underscores the need for long-term medical and psychological support for recovered patients. These findings emphasize the importance of comprehensive post-recovery **care**, including rehabilitation programs, cognitive interventions, and mental health services to support individuals in regaining their overall well-being. Future research should focus on developing targeted interventions to address these long-term effects and improve the quality of life for COVID-19 survivors. These interventions such as cognitive rehabilitation strategies, such as attention and memory training, for individuals with lingering neurocognitive impairments.

## Author contributions

**Conceptualization:** Sawsan Abuhammad, Karem H. Alzoubi, Shaher Hamaideh, Maryam El-zubi.

**Data curation:** Basheer Khasawneh.

**Formal analysis:** Sawsan Abuhammad, Karem H. Alzoubi, Shaher Hamaideh, Maryam El-zubi.

**Funding acquisition:** Sawsan Abuhammad, Karem H. Alzoubi, Omar F. Khabour, Basheer Khasawneh, Maryam El-zubi.

**Investigation:** Sawsan Abuhammad, Omar F. Khabour, Shaher Hamaideh, Basheer Khasawneh, Maryam El-zubi.

**Methodology:** Karem H. Alzoubi, Shaher Hamaideh, Basheer Khasawneh.

**Project administration:** Sawsan Abuhammad, Omar F. Khabour, Basheer Khasawneh, Maryam El-zubi.

**Resources:** Karem H. Alzoubi, Omar F. Khabour, Basheer Khasawneh, Maryam El-zubi.

**Software:** Sawsan Abuhammad, Karem H. Alzoubi, Omar F. Khabour, Maryam El-zubi.

**Supervision:** Sawsan Abuhammad, Karem H. Alzoubi, Omar F. Khabour, Shaher Hamaideh, Basheer Khasawneh, Maryam El-zubi.

**Validation:** Sawsan Abuhammad, Omar F. Khabour, Shaher Hamaideh, Basheer Khasawneh, Maryam El-zubi.

**Visualization:** Sawsan Abuhammad, Karem H. Alzoubi, Omar F. Khabour, Shaher Hamaideh, Basheer Khasawneh, Maryam El-zubi.

**Writing – original draft:** Sawsan Abuhammad, Omar F. Khabour, Shaher Hamaideh, Maryam El-zubi.

**Writing – review & editing:** Omar F. Khabour, Shaher Hamaideh, Basheer Khasawneh, Maryam El-zubi.

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
