## [Decision Letter · Decision Letter 0]

24 Apr 2024

PONE-D-24-05191COVID-19 Recovered Patients' Physical and Mental Health Well-Being:  Phenomenological StudyPLOS ONE

Dear Dr.** **Abuhammad,

Thank you for submitting your manuscript to PLOS ONE. After careful consideration, we feel that it has merit but does not fully meet PLOS ONE’s publication criteria as it currently stands. Therefore, we invite you to submit a revised version of the manuscript that addresses the points raised during the review process.

**ACADEMIC **

Ensure that you use line numbering throughout the manuscript to ensure easy review of your manuscript. This will accord reviewers the ease of stating where corrections need to be effected. The whole manuscript needs to be reviewed for proper presentation.

Title: the title of the setting should contain the study area/ setting, what type of recovered patients were interviewed- hospital staff? Or a certain community? And the country of the study. Instead of phenomenological study, I suggest that the heading should be a qualitative study.

Abstract

The abstract should contain, Introduction, Methods, Results and Conclusion.

Introduction should have a statement on the subject matter before stating the aim/objective of the study.

Methods; unstructured interview protocol? Does it mean that you did not use an interview guide / a checklist / a semi-structured interview guide to guide the discussions? Didn’t you have questions which you used to aid in meeting the aim of your study? State your sampling method that was used – purposive? Snowball?

State the method you used in analyzing your data? Phenomenological method? Inductive method of coding? Reflexivity?

Introduction

The first paragraph should be on the global trend of the post COVID-19 traumatic experiences of patients that had moderate to severe COVID-19 infection. Followed by the continental post COVID-19 traumatic experiences of patients that had moderate to severe COVID-19 infection and finally the country where this study was conducted.

The statement below should be part of the last paragraph of the introduction as it relates to the relevance of why the study was conducted.

“The study of post-severe infection health is not merely an academic endeavor; it is an urgent call to understand, address, and alleviate the intricate and enduring challenges faced by survivors who have traversed the often arduous journey of severe COVID-19(2, 7).”

Please, ensure that the introduction is constructed in such a way that there is good flow of information, relevant information and concise.

Methods

Start with the study setting where the information was collected. We need the city or states were the information was collected and the country, not just stating that the interview was house-to-house or preferred setting suggested.

State clearly the eligibility criteria of your study.

What type of sampling method that was used in the selection of study participants?

How did you ensure that bias is minimal in your study and the validity and reliability of the study? Reflexivity? Inductive method of coding?

Did you try to enhance the interview by making statements like- tell us more about your experience or can you please throw more light or explain more on the statement you made? Please, do state how you circumvented your questions in the data collection section.

What was the approximate timing of each participant during the interview? Did you train researchers who conducted the interviews?

How many people were involved in the data collection processes?

Was there at any point you reached “data saturation point” during your interview? Kindly state it in the METHODS section and if you did not reach data saturation point, do ensure that you state it under the limitations of your study in the DISCUSSION section. Since “data saturation point” indicates that there was no new information that was generated during the interview.

Ensure that the interview guide questions are also stated under the methodology.

In what language was the interview conducted?

Who was responsible for the translation? Was he/she an expert in the language? Clearly state it in  the methodology.

Your data collection should be separated from ETHICAL CONSIDERATIONS/APPROVAL.

Under ethical considerations you can state that written informed consent was obtained from participants that agreed to participate in the study after thorough explanation of what the study entails including stating of risk involved if any, the scope of the study and their ability to withdraw at any point of the study if the study was not convenient for them. The ethical approval of the study should also be stated under this section.

Results

The demographic characteristics also contained some analysed quantitative descriptive analysis of the participants’ experience of the symptoms of the COVID-19 infection. Should this be part of the demographic characteristics?

What about the age group of those that were interviewed (participants)? Do state the age range of the participants.

You need to insert a table that in this section that contains; the themes, subthemes, codes and the frequency of code generated by each participant

Under the theme generated for example PHYSICAL HEALTH; were respiratory problems, joint and muscle pain, changes in activity level, and worsening of previous health problems the subthemes that led to the formation of the afore mentioned theme? Please endeavor to state some of the statements from the interviews conducted that fell under this categories.

Which categories led to the formation of theme **respiratory problems**? Please, indicate and write the experiences of the participants under each category

Four themes were generated from the study but it seems like up to 12 themes were covered in the result section. Please ensure that you clearly state the themes and subthemes while at the same time directing the statements under each subtheme or categories generated as earlier stated.

Discussion

There is need for a more elaborative description of your findings in relation to other qualitative/quantitative studies that signified such experiences by patients that recovered from COVID-19.

It is expected that your discussion should follow the presentation of your result. Use a paragraph for each theme that was developed and discuss sequentially as presented in the result section.

Finally, you should high light the limitations of your study as the last paragraph

I suggest that you use other published qualitative articles as guide and also use the COREQ checklist for qualitative studies.

We look forward to receiving your revised manuscript.

Kind regards,

Ayi Vandi Kwaghe, D.V.M., M.V.Sc., P.G.D.E. Ph.D., MPH, FETP

Academic Editor

PLOS ONE

Journal Requirements:

3. In the online submission form, you indicated that [Data will be avaliable upon reasonable request. It is include recordings and sensitive data]. 

Reviewers' comments:

Reviewer's Responses to Questions

**Comments to the Author**

1. Is the manuscript technically sound, and do the data support the conclusions?

Reviewer #1: Yes

Reviewer #2: Yes

2. Has the statistical analysis been performed appropriately and rigorously? 

Reviewer #1: N/A

Reviewer #2: Yes

3. Have the authors made all data underlying the findings in their manuscript fully available?

Reviewer #1: Yes

Reviewer #2: Yes

4. Is the manuscript presented in an intelligible fashion and written in standard English?

Reviewer #1: No

Reviewer #2: Yes

5. Review Comments to the Author

Reviewer #1: This study included useful information regarding post Covid syndrome which involved millions of people worldwide. familiarity with the finding of this study will help nurses to consider them during their nursing care

Reviewer #2: I have read the referred article with keen interest. The information is interesting and innovative; conclusion section is interesting and authors can improve it further. I am recommending authors to do a little more work and add latest literate to support the study. The authors need to improve results section. The level of English is good and smooth, e.g., the language standard, specifically the grammar, of sufficient quality to meet scientific merit for publication. However, I suggest authors to double check for language quality. Describe scientific contribution of the study to the existing body of knowledge. I endorse this manuscript after minor revision as suggested. The topic is interesting and worthy of attention. The methodology is adequate and the conclusions are consistent with the reported data. The manuscript can be improved by expanding the references and citing some recently published articles on this topic.

Authors should consider the following recommendations:

- I recommend further improving the references by citing some of these recent studies on the topic:

Naeem, B., Aqeel, M., & de Almeida Santos, Z. (2021). Marital conflict, self-silencing, dissociation, and depression in married madrassa and non-madrassa women: a multilevel mediating model. Nature-Nurture Journal of Psychology, 1(2), 1-11.

Naeem, B., & Chaman, A. The Association of Adverse Self-Silencing and Marital Conflict with Symptoms of Depression and Dissociation in Married Madrassa and Non-Madrassa Women: A Cross-sectional Study.

Naeem, B. Nurturing the Soul: A Psychometric Analysis of the Spiritual Intelligence Inventory in Married Madrassa and Non-Madrassa Women.

Saif, J., Rohail, D. I., & Aqeel, M. (2021). Quality of Life, Coping Strategies, and Psychological Distress in Women with Primary and Secondary Infertility; A Mediating Model . Nature-Nurture Journal of Psychology, 1(1 SE-), 8–17.

Naeem, B., Aqeel, M., & de Almeida Santos, Z. (2021). Marital Conflict, Self-Silencing, Dissociation, and Depression in Married Madrassa and Non-Madrassa Women: A Multilevel Mediating Model. Nature-Nurture Journal of Psychology, 1(2), 1–11

Hafsa, S., Aqeel, M., & Shuja, K. H. (2021). The Moderating Role of Emotional Intelligence between Inter-Parental Conflicts and Loneliness in Male and Female Adolescents. Nature-Nurture Journal of Psychology, 1(1 SE-), 38–48

Rashid, A., Aqeel, M., Malik, D. B., & Salim, D. S. (2021). The Prevalence of Psychiatric Disorders in Breast Cancer Patients; A Cross-Sectional Study of Breast Cancer Patients Experience in Pakistan. Nature-Nurture Journal of Psychology, 1(1 SE-), 1–7. https://thenaturenurture.org/index.php/psychology/article/view/1

Sarfraz, R., Aqeel, M., Lactao, D. J., & Khan, D. S. (2021). Coping Strategies, Pain Severity, Pain Anxiety, Depression, Positive and Negative Affect in Osteoarthritis Patients; A Mediating and Moderating Model . Nature-Nurture Journal of Psychology, 1(1 SE-), 18–28. https://thenaturenurture.org/index.php/psychology/article/view/8

Aqeel, M., Nisar, H. H., Rehna, T., & Ahmed, A. (2021). Self-harm behaviour, psychopathological distress and suicidal ideation in normal and deliberate self-harm outpatient’s adults. Journal of the Pakistan Medical Association, 71(9), 2143-2147.

Aqeel, M., Rohail, I., Ahsan, S., Ahmed, A., & Saad, M. B. (2017). Moderating role of the forgiveness between vengeance and aggression in Pakistani murderers. Wulfenia Journal, 24(3), 269-283.

Aqeel, M., Jami, H., & Ahmed, A. (2017). Translation, adaptation, and cross-language validation of student: thinking about my homework scale (STP). International Journal of Human Rights in Healthcare, 10(5), 296-309.

6. PLOS authors have the option to publish the peer review history of their article (what does this mean?). If published, this will include your full peer review and any attached files.

Reviewer #1: **Yes: **Alireza Nikbakht Nasrabadi

Reviewer #2: **Yes: **Dr.Muhammad Aqeel

---

## [Decision Letter · Decision Letter 1]

18 Feb 2025

PONE-D-24-05191R1COVID-19 Survivor Parents' Physical and Mental health well-being: A Qualitative Study from JordanPLOS ONE

Dear Dr. Abuhammad,

Thank you for submitting your manuscript to PLOS ONE. After careful consideration, we feel that it has merit but does not fully meet PLOS ONE’s publication criteria as it currently stands. Therefore, we invite you to submit a revised version of the manuscript that addresses the points raised during the review process.

We look forward to receiving your revised manuscript.

Kind regards,

Ayi Vandi Kwaghe, D.V.M., M.V.Sc., P.G.D.E. Ph.D., MPH, FETP

Academic Editor

PLOS ONE

Journal Requirements:

Reviewers' comments:

Reviewer's Responses to Questions

**Comments to the Author**

1. If the authors have adequately addressed your comments raised in a previous round of review and you feel that this manuscript is now acceptable for publication, you may indicate that here to bypass the “Comments to the Author” section, enter your conflict of interest statement in the “Confidential to Editor” section, and submit your "Accept" recommendation.

Reviewer #3: (No Response)

Reviewer #4: (No Response)

Reviewer #5: (No Response)

2. Is the manuscript technically sound, and do the data support the conclusions?

Reviewer #3: Partly

Reviewer #4: Partly

Reviewer #5: Yes

3. Has the statistical analysis been performed appropriately and rigorously? 

Reviewer #3: N/A

Reviewer #4: Yes

Reviewer #5: Yes

4. Have the authors made all data underlying the findings in their manuscript fully available?

Reviewer #3: Yes

Reviewer #4: Yes

Reviewer #5: Yes

5. Is the manuscript presented in an intelligible fashion and written in standard English?

Reviewer #3: Yes

Reviewer #4: No

Reviewer #5: No

6. Review Comments to the Author

Reviewer #3: Dear authors

Thank you for your efforts to do this research. Below are some comments and suggestions to help improve the clarity and impact of your study.

The authors discuss the COVID-19 Survivor Parents' Physical and Mental health well-being; however, there is no clear explanation as to why parents specifically were chosen as the focus of the study. Furthermore, simply considering parents as a group is insufficient. The term "parents" is broad and encompasses various subgroups. It is not clear what criteria were used to define parents in the study. For instance, the criteria of being hospitalized or receiving oxygen therapy are not distinguishing enough. The study could benefit from distinguishing among different types of parents, such as those with healthy children, those with children who have chronic conditions, those who lost a child to COVID-19, and those with children who require transplants. Each of these subgroups would likely have different experiences.

The study appears to treat all COVID-19 survivor parents as a homogeneous group, as if investigating the experiences of ordinary survivors without considering the diversity among them. Additionally, in the introduction, the authors state, "Many survivor parents of COVID-19 suffer from lingering physical symptoms, often referred to post-COVID-19(14, 15)". However, the references reviewed did not specifically address parents.

Reviewer #4: The paper presents the findings of an intriguing qualitative study on the experience of patients who have recovered from Coronavirus Disease 2019 with respect to their physical and psychological quality of life. The topic is both timely and of significant interest; however, it is recommended that responses address the methodology and the use of excessive diagnostic terminology, and that the paper be proofread by native speakers to enhance its scientific English.

Reviewer #5: Note: I reviewed the first manuscript in the submission. There was a “revised” manuscript at the end that was not reviewed or compared to the first.

This qualitative study examined the occurrence of persistent post-COVID-19 health issues using unstructured interviews in patients who had severe COVID-19.

The study was sufficiently methodically sound to ensure the results have low bias and are valid across raters.

The discussion/conclusion could be improved; please explicitly tell what you found, what you expected, any differences between the two and point out any new unexpected themes. These aspects are somewhat present, but please work on making them more explicit.

There are a large number of small grammatical and language issues that I suspect are a result of the authors not having English as their first language. These should be fixed in order to make the manuscript appear professional. There are so many, I stopped trying to note each one. I recommend hiring or otherwise utilizing a writer experienced scientific writing in English to do a full pass to fix these issues. Here is the incomplete list of items I identified before I stopped, use the search function to locate the problem areas I identified:

• Introduction of abstract. Break into 2 or more sentences. Opening sentence is not clear, I think you mean “Understanding Post-severe infection health…”. Revise.

• In the abstract results, the sentence, “The cognitive health subheading include subheadings such as loss of memory and diminish ability of focusing and changes in cognitive abilities.” should only mention subheadings once. Revise

• Do not use colloquial language. Here are some phrases that need to be replaced with precise scientific language.

o “sobering light” – being sober or not is irrelevant.

o “treacherous path”. --- they are not literally walking on a path.

o “grapples” / “grappling --- I don’t think wrestling is involved

o “illuminating”

o “forge a path”

o “Embark on a journey”

• Replace “This subject studying” with “This study”. Pg. 10

• In, “Many survivor parents of COVID-19 suffer from

• lingering physical symptoms, often referred to post-COVID-19(14, 15).” The last part of this sentence is not clear.

• “The impact of COVID-19 pandemic is profound…”. Should be “The impact of THE COVID-19”

• “and all survivors that includes parents showed suffering from mental health issues”. Remove “that includes parents” I’m not sure why this is in there.

My recommendation is that this study is suitable for publication if the above issues are addressed satisfactorily after revision.

7. PLOS authors have the option to publish the peer review history of their article (what does this mean?). If published, this will include your full peer review and any attached files.

Reviewer #3: No

Reviewer #4: **Yes: **Silvia Costanzo, Psyhologist-Psychotherapist and Researcher

Reviewer #5: No

---

## [Decision Letter · Decision Letter 2]

31 Mar 2025

PONE-D-24-05191R2Physical and Mental Health Well-Being of COVID-19 Recovered Patients: A Phenomenological StudyPLOS ONE

Dear Dr. Abuhammad,

Thank you for submitting your manuscript to PLOS ONE. After careful consideration, we feel that it has merit but does not fully meet PLOS ONE’s publication criteria as it currently stands. Therefore, we invite you to submit a revised version of the manuscript that addresses the points raised during the review process.

We look forward to receiving your revised manuscript.

Kind regards,

Ayi Vandi Kwaghe, D.V.M., M.V.Sc., P.G.D.E. Ph.D., MPH, FETP

Academic Editor

PLOS ONE

Journal Requirements:

Reviewers' comments:

Reviewer's Responses to Questions

**Comments to the Author**

1. If the authors have adequately addressed your comments raised in a previous round of review and you feel that this manuscript is now acceptable for publication, you may indicate that here to bypass the “Comments to the Author” section, enter your conflict of interest statement in the “Confidential to Editor” section, and submit your "Accept" recommendation.

Reviewer #4: All comments have been addressed

Reviewer #5: All comments have been addressed

Reviewer #6: (No Response)

2. Is the manuscript technically sound, and do the data support the conclusions?

Reviewer #4: Yes

Reviewer #5: Yes

Reviewer #6: Yes

3. Has the statistical analysis been performed appropriately and rigorously? 

Reviewer #4: Yes

Reviewer #5: Yes

Reviewer #6: N/A

4. Have the authors made all data underlying the findings in their manuscript fully available?

Reviewer #4: Yes

Reviewer #5: No

Reviewer #6: Yes

5. Is the manuscript presented in an intelligible fashion and written in standard English?

Reviewer #4: Yes

Reviewer #5: Yes

Reviewer #6: Yes

6. Review Comments to the Author

Reviewer #4: The authors have correctly revised the paper in accordance with the reviewer's instructions. The text has undergone a thorough proofreading process, resulting in enhanced fluency.

Reviewer #5: The manuscript language is very much improved. Thanks for your excellent contribution and congratulations on publishing!

Reviewer #6: The study addresses an important topic. The aim of the study is clearly stated, and the focus on both physical and mental health aspects is crucial for understanding the holistic recovery of individuals post-infection.

The use of a qualitative approach, particularly unstructured interviews, is appropriate for exploring participants' experiences in-depth. This method allows for a nuanced understanding of the subjective experiences of individuals recovering from COVID-19, which is critical for informing both clinical practice and future research.

However, I have a few suggestions and questions regarding the study's methodology:

Subject Selection and Sampling:

Could the authors specify how many hospitals were selected for participant recruitment and how many patients were selected from each hospital? This detail will help readers understand the scope and diversity of the sample.

Sampling Method and Sample Size:

It would be beneficial if the authors could describe the sampling method in more detail. Additionally, it would be helpful to explain why a sample size of 30 participants was chosen.

Use of Qualitative Software:

If a qualitative software was used to analyze the data, the authors should provide more details on how it was utilized (e.g., for coding or data organization). Additionally, citing the software's references and giving a brief explanation of its role in the analysis would enhance the transparency of the research process.

Participant Compensation:

Were participants compensated for their participation in the study? If so, it would be helpful to mention this information to ensure transparency and address ethical considerations. If not, an explanation of the rationale for not compensating participants would be appreciated.

Conclusion:

The conclusion would benefit from a more detailed discussion of specific recommendations

7. PLOS authors have the option to publish the peer review history of their article (what does this mean?). If published, this will include your full peer review and any attached files.

Reviewer #4: **Yes: **Dr. Silvia Costanzo,

Psychologist-Psychotherapist and Researcher

IRCCS Istituto Tumori Giovanni Paolo II, Bari

Reviewer #5: No

Reviewer #6: No

---

## [Editor Report · Decision Letter 3]

25 Apr 2025

Physical and Mental Health Well-Being of COVID-19 Recovered Patients: A Phenomenological Study

PONE-D-24-05191R3

Dear Dr. **Abuhammad**,

We’re pleased to inform you that your manuscript has been judged scientifically suitable for publication and will be formally accepted for publication once it meets all outstanding technical requirements.

Kind regards,

Ayi Vandi Kwaghe, D.V.M., M.V.Sc., P.G.D.E. Ph.D., MPH, FETP

Academic Editor

PLOS ONE
---

## [Editor Report · Acceptance letter]

PONE-D-24-05191R3

PLOS ONE

Dear Dr. Abuhammad,

I'm pleased to inform you that your manuscript has been deemed suitable for publication in PLOS ONE. Congratulations! Your manuscript is now being handed over to our production team.

Kind regards,

on behalf of

Dr. Ayi Vandi Kwaghe

Academic Editor

PLOS ONE